# Synergistic Antimicrobial Effect of Colistin in Combination with Econazole against Multidrug-Resistant *Acinetobacter baumannii* and Its Persisters

Miaomiao Xie,[a] Kaichao Chen,[a] Edward Wai-Chi Chan,[a,b] Sheng Chen[a]

[a]Department of Infectious Diseases and Public Health, Jockey Club College of Veterinary Medicine and Life Sciences, City University of Hong Kong, Kowloon, Hong Kong
[b]State Key Lab of Chemical Biology and Drug Discovery, Department of Applied Biology and Chemical Technology, The Hong Kong Polytechnic University, Kowloon, Hong Kong

**ABSTRACT** Colistin is a last-line antibiotic which acts by causing membrane permeabilization in Gram-negative bacteria. However, its clinical value has been limited by its toxicity and the emergence of resistant organisms. In this study, we showed that econazole and colistin can act synergistically to produce a strong antimicrobial effect sufficient for eradication of starvation-induced tolerant and multidrug-resistant populations of *Acinetobacter baumannii*, a notorious pathogen causing recalcitrant infections, both *in vitro* and in mouse infection models. Investigation of the underlying mechanism showed that, while colistin disrupts the membrane structure, econazole causes the dissipation of proton motive force, eliciting a vicious cycle of membrane structural damages and disruption of membrane protein functions, and eventually cell death. This drug combination therefore achieves our goal of using a much smaller dosage of colistin to produce a much stronger antimicrobial effect to tackle the problems of toxicity and resistance associated with colistin usage.

**IMPORTANCE** Findings described in this study constitute concrete evidence that it is possible to significantly enhance the antimicrobial activity of colistin by using an antifungal drug, econazole, as a colistin adjuvant. We showed that this drug combination can kill not only multidrug-resistant *A. baumannii* but also the tolerant subpopulation of such strains known as persisters, which may cause chronic and recurrent infections in clinical settings. The synergistic killing effect of the econazole and colistin combination was also observable in mouse infection models at a very low concentration, suggesting that such a drug combination has high potential to be used clinically. Findings in this study therefore have important implications for enhancing its clinical application potential as well as developing new approaches to enhance treatment effectiveness and reduce suffering in patients.

**KEYWORDS** *Acinetobacter baumannii*, persisters, synergistic antimicrobial effect, econazole, colistin

A*cinetobacter baumannii* is one of the most important nosocomial pathogens (1), especially in intensive care units, as it can colonize different organs of patients and survive for a long period of time on abiotic surfaces (2), causing various infectious diseases including bacteremia, ventilator-associated pneumonia, catheter-related infections, meningitis, peritonitis, and urinary infections (3–5). Multidrug-resistant (MDR) *A. baumannii* has become a critical health concern in the last decade (6, 7), since infections caused by such strains are often associated with high nosocomial morbidity and mortality (4, 8). In addition to multidrug resistance, *A. baumannii* also exhibits multidrug tolerance, a phenomenon in which a subpopulation known as persisters does not respond to antimicrobial action and may cause a range of chronic infections (9, 10). Unlike the resistant organisms, persisters do not grow in the presence of antibiotics but instead go into a state of dormancy which

Address correspondence to Sheng Chen, shechen@cityu.edu.hk.

The authors declare no conflict of interest.

renders the cell unresponsive to antibiotics (11). Recent studies reported that expression of the persister phenotype can be triggered by starvation, oxidative stress, and quorum sensing (12, 13). In particular, bacteria become highly tolerant to antimicrobial agents when nutrients become limited. The inactivation of antibiotic targets due to starvation-induced growth arrest is considered a crucial mechanism of tolerance (14). Our laboratory recently showed that bacteria actually need to actively maintain proton motive force (PMF) in order to express phenotypic tolerance (15). It should be noted that starvation often contributes to antibiotic tolerance during infections, since nutrients become unavailable when bacteria are isolated by the host defense system, or when depleted by proliferating bacteria (16). We therefore hypothesize that suppressing PMF may lead to reduced viability of the infecting agents.

Present therapeutic options for infections caused by *A. baumannii* are limited. Meropenem, tigecycline, minocycline, and amikacin have been adopted for treatment of *A. baumannii* infections (17). Polymyxins, which were previously prohibited from clinical usage due to their nephrotoxicity and neurotoxicity, are now being used as the last-resort agent for treatment of multidrug-resistant *A. baumannii* infections (18). However, despite the high *in vitro* activity of polymyxins, a number of factors, including limited clinical efficacy and emergence of antimicrobial resistance during treatment, have prompted clinicians to use combination therapy to combat multidrug-resistant *A. baumannii* infections (19, 20). Treatment with a combination of antibiotics has also been proposed as an effective way to eradicate persisters that cause chronic and recurrent infections (21, 22). However, treatment failure is common, prompting a need to develop more effective combinations of antibiotics and/or other agents to improve antimicrobial efficacy (23). Econazole is an imidazole-type antifungal drug used to treat systemic and topical fungal infections (24). In a recent study, we found that it could act in synergism with colistin against colistin-resistant *Enterobacteriaceae* (25). In this work, we investigated the antitolerance potential of econazole and colistin combination, using *A. baumannii* as a model organism. Our preliminary data show that this drug combination exhibited strong antimicrobial activity against these difficult-to-treat *A. baumannii* infections. We have gathered sufficient evidence which confirms that this drug combination can effectively eradicate both antibiotic-resistant and -tolerant *A. baumannii* populations.

## RESULTS

**Econazole and colistin combination exhibits synergistic antimicrobial effects on multidrug-resistant *A. baumannii*.** To evaluate the potential antimicrobial efficacy of econazole and its ability to enhance colistin activity, checkerboard assays were performed. We found that econazole potentiated colistin activity against both susceptible and multidrug-resistant *A. baumannii* strains with fractional inhibitory concentration (FIC) indexes (FICIs) of 0.1875 and 0.25 for strains ATCC 17978 and CPC35, respectively (Fig. 1). The colistin MIC of *A. baumannii* ATCC 17978 was reduced from 4 $\mu$g/mL to 0.25 $\mu$g/mL when measured in combination with 4 $\mu$g/mL econazole, with further decrease to 0.06 $\mu$g/mL observed in combination with 16 $\mu$g/mL econazole (Fig. 1a). Furthermore, combination of econazole and colistin also exhibited a strong synergistic antimicrobial effect on multidrug-resistant (including carbapenem-resistant) *A. baumannii* CPC35, with results showing that the colistin MIC was reduced from 2 $\mu$g/mL to 0.25 $\mu$g/mL when used in combination with 4 $\mu$g/mL econazole. Moreover, the colistin MIC of strain CPC35 was significantly decreased to 0.015 $\mu$g/mL (a 128-fold decrease) when econazole was present at a concentration of 16 $\mu$g/mL (Fig. 1b), suggesting that econazole exhibited a dose-dependent enhancement effect on the ability of colistin to kill both susceptible and multidrug-resistant *A. baumannii* strains.

**Econazole and colistin combination kills multidrug-resistant *A. baumannii* and its persisters *in vitro*.** To further investigate the synergistic bactericidal effect of econazole and colistin combination, time-dependent killing assays were conducted on bacteria of different physiological statuses, including *A. baumannii* bacteria in exponential growth phase and those subjected to starvation for 24 h. In this experiment, we found that *A. baumannii* ATCC 17978 at exponential growth phase could not be killed or inhibited by treatment with 8 $\mu$g/mL econazole alone; treatment with 2 $\mu$g/mL colistin alone caused only a

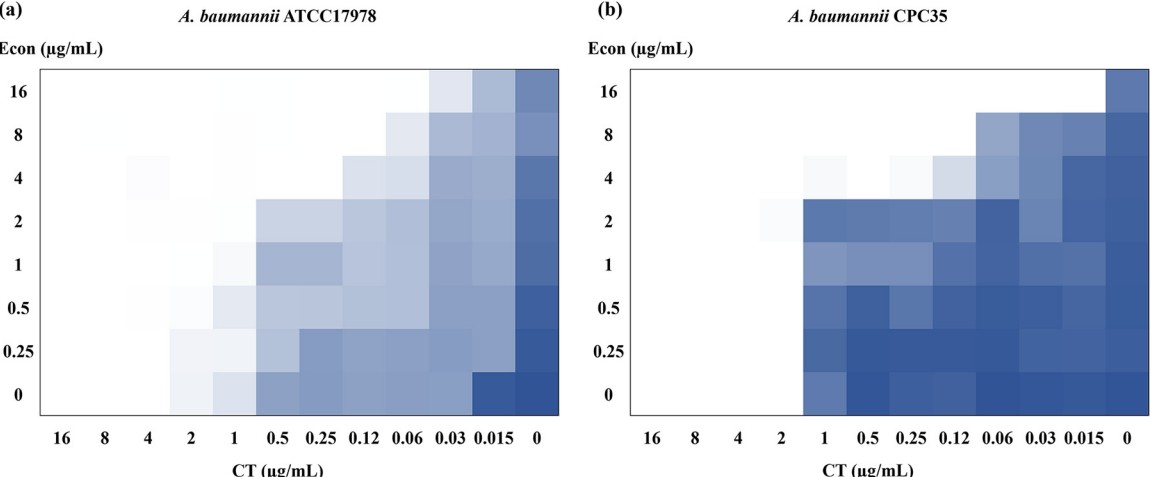

**FIG 1** Checkerboard assays of synergistic antimicrobial activity of colistin and econazole combinations against susceptible *A. baumannii* ATCC 17978 (a) and multidrug-resistant *A. baumannii* CPC35 (b). After 18 h of incubation, the absorbance of the bacterial culture at 600 nm was shown. Dark blue regions represent higher cell density. Data represent the mean $OD_{600}$ from three biological replicates. Econ, econazole; CT, colistin.

slight reduction in bacterial population size. However, upon combined treatment with 2 $\mu$g/mL colistin and 8 $\mu$g/mL econazole, the strain ATCC 17978 population was completely eradicated at 6 h (Fig. 2a). A synergistic bactericidal effect of econazole and colistin combination could also be observed in exponentially growing, multidrug-resistant *A. baumannii* CPC35; importantly, complete eradication of the entire population of this multidrug-resistant strain was observed within 1 h upon treatment with a combination of 8 $\mu$g/mL econazole and 2 $\mu$g/mL colistin. Complete eradication was also achieved at 4 h upon treatment with 4 $\mu$g/mL econazole and 2 $\mu$g/mL colistin (Fig. 2b). For a starvation-induced tolerant population of *A. baumannii* (both strain ATCC 17978 and strain CPC35), treatment

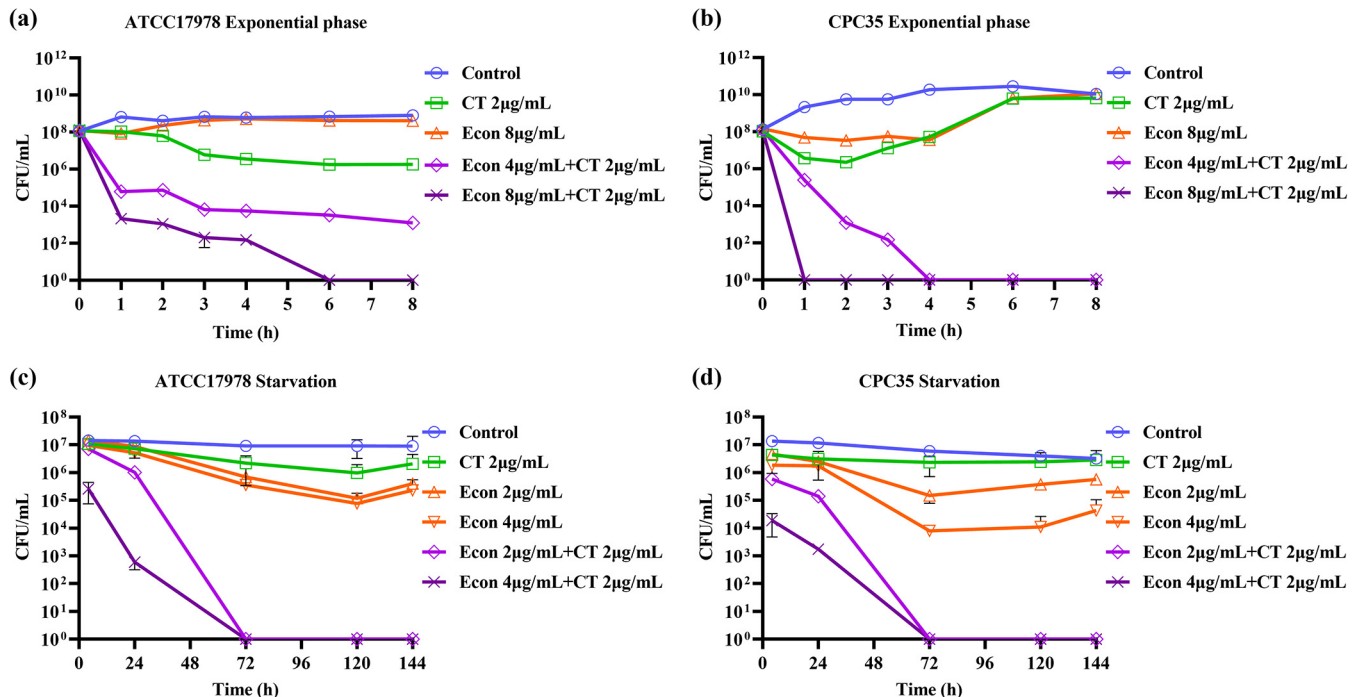

**FIG 2** Time-dependent killing assays of *A. baumannii*. (a and b) *A. baumannii* ATCC 17978 (a) and CPC35 (b) were incubated to exponential phase and challenged with econazole, colistin, or combinations of the two compounds. (c and d) *A. baumannii* ATCC 17978 (c) and CPC35 (d) were subjected to starvation for 24 h and challenged with econazole, colistin, or combinations of the two agents for 144 h. Data were shown as mean ± standard deviation from three independent experiments.

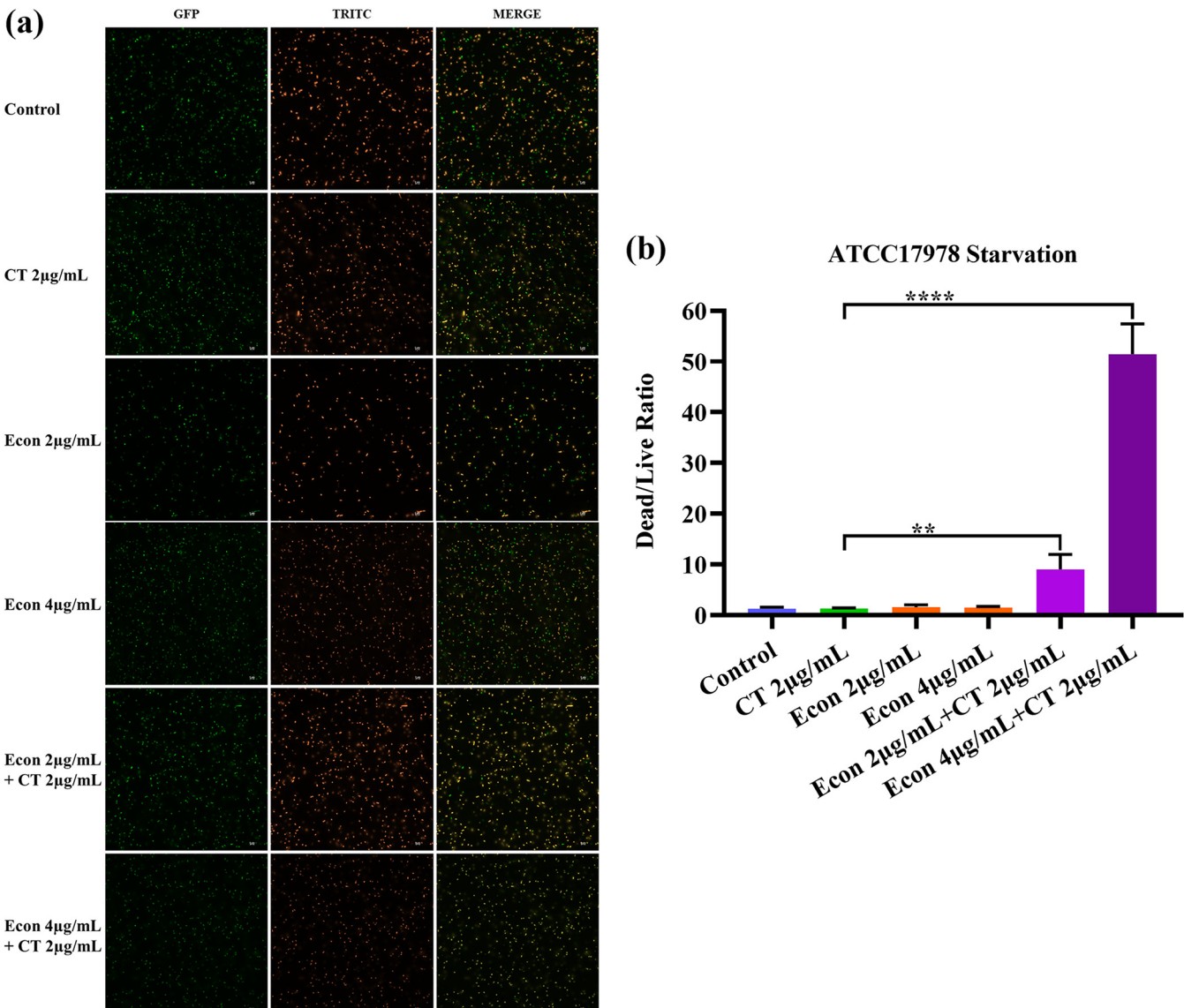

**FIG 3** LIVE/DEAD staining to depict synergistic bactericidal effect of colistin and econazole combination on starvation-induced tolerant *A. baumannii* ATCC 17978. (a) Microscopic images of *A. baumannii* ATCC 17978 subjected to starvation and drug treatments, followed by LIVE/DEAD staining. Strain ATCC 17978 was subjected to starvation for 24 h and challenged with colistin, econazole, or combinations of the two for 4 h. (b) Dead/live ratio after strain ATCC 17978 was subjected to starvation and treatment with different agents for 4 h. Data were analyzed using one-way ANOVA and the Tukey test. Significant differences were observed between bacteria treated with 2 $\mu$g/mL colistin alone and combination of 2 $\mu$g/mL colistin and 2 $\mu$g/mL econazole (**, $P$ = 0.0093) and combination of 2 $\mu$g/mL colistin and 4 $\mu$g/mL econazole (****, $P$ < 0.0001).

with 2 $\mu$g/mL colistin alone or 4 $\mu$g/mL econazole alone for 144 h could not effectively kill the entire tolerant bacterial populations. However, combined treatment with 2 $\mu$g/mL colistin and 2 $\mu$g/mL econazole resulted in complete eradication of the tolerant bacterial population of these two strains at 72 h (Fig. 2c and d). These results suggested that econazole significantly enhanced the bactericidal effect of colistin on both exponential and starvation-induced tolerant *A. baumannii* strains.

A synergistic bactericidal effect of econazole and colistin combination on a starvation-induced tolerant population of *A. baumannii* ATCC 17978 was also illustrated by LIVE/DEAD staining (Fig. 3a). Upon starvation for 24 h in saline, treatment with 2 $\mu$g/mL colistin alone or 4 $\mu$g/mL econazole alone could not kill the tolerant *A. baumannii* ATCC 17978 population, in which death at a proportion similar to that in the control group was observed. However, combined treatment with 2 $\mu$g/mL colistin and 2 $\mu$g/mL econazole

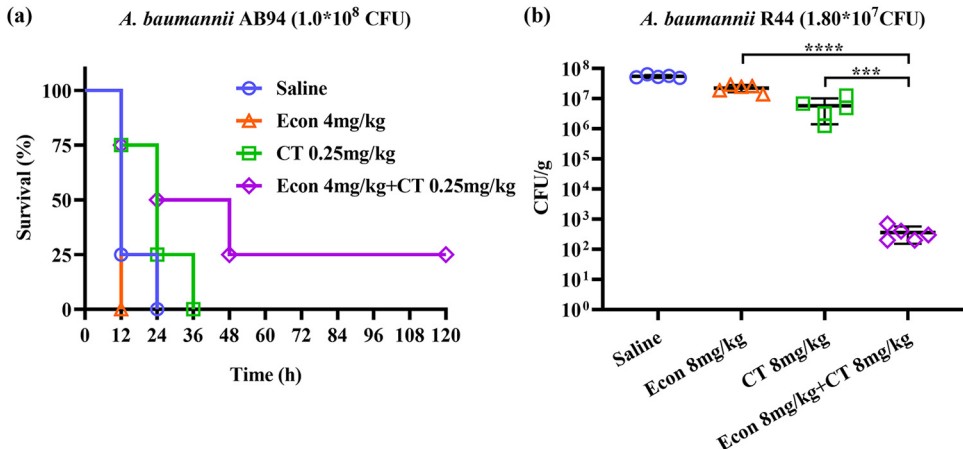

**FIG 4** Synergistic antimicrobial activity of econazole and colistin combination in mouse infection models. (a) Mouse sepsis model. Neutropenic ICR mice ($n = 8$) were infected with $1.0 \times 10^8$ CFU of clinical carbapenem-resistant *A. baumannii* AB94 by intravenous injection and subjected to different treatments postinfection. Treatments included saline, 4 mg/kg econazole, 0.25 mg/kg colistin, and a combination of econazole (4 mg/kg) and colistin (0.25 mg/kg). Survival of mice in each group was observed and recorded for 120 h postinfection ($P = 0.0396$). Statistical analysis was performed by log rank (Mantel-Cox) test. (b) Mouse thigh infection model. Neutropenic ICR mice ($n = 5$) were infected with $1.80 \times 10^7$ CFU of clinical carbapenem- and colistin-resistant *A. baumannii* R44 in each thigh. After infection for 24 h, mice were subjected to antimicrobial treatments including 8 mg/kg econazole, 8 mg/kg colistin, and a combination of econazole (8 mg/kg) and colistin (8 mg/kg), at 12-h intervals for a total of 72 h. Saline was included as a no-treatment control. Mice were then euthanized, and bacterial loads in both thighs were determined. Data were statistically analyzed with one-way ANOVA and the Tukey test (****, $P < 0.0001$; ***, $P = 0.0003$).

resulted in death of 89.46% of the population (dead/live ratio, 9.05), with further increase in percentage of dead populations (97.73%; dead/live ratio, 46.4) being observed when challenged with 2 $\mu$g/mL colistin and 4 $\mu$g/mL econazole (Fig. 3a and b).

**Econazole and colistin combination kills *A. baumannii* in mouse infection models.** The i*n vivo* synergistic bactericidal effect of econazole and colistin combination was evaluated in a mouse sepsis model, in which mice intravenously infected with $1.0 \times 10^8$ CFU of carbapenem-resistant *A. baumannii* AB94 died within 24 h postinfection when treated with saline or 4 mg of econazole alone per kg of body weight (Fig. 4a). When treated with 0.25 mg/kg colistin, 75% of infected mice died at 24 h postinfection, and the remaining 25% of infected mice died at 36 h postinfection. However, treatment with a combination of 4 mg/kg econazole and 0.25 mg/kg colistin could effectively rescue 25% of infected mice within 120 h postinfection ($P = 0.0396$). In a mouse thigh infection model in which mice were infected by $1.80 \times 10^7$ CFU of carbapenem- and colistin-resistant *A. baumannii* R44, combined treatment with 8 mg/kg econazole and 8 mg/kg colistin resulted in significant decrease in bacterial loads of this colistin-resistant strain in infected thighs compared to treatment with econazole or colistin alone ($P < 0.0001$ and $P = 0.0003$, respectively) (Fig. 4b). These data indicated that econazole could effectively potentiate colistin activity *in vivo* in treatment of infections caused by both carbapenem-resistant and colistin-resistant *A. baumannii* strains.

**Econazole enhances the membrane-damaging effect of colistin.** Colistin expresses its antimicrobial activity against Gram-negative bacteria by specifically interacting with lipopolysaccharide in the bacterial outer membrane. We hypothesized that the reason why econazole could enhance the antimicrobial effect of colistin was because econazole could enhance its ability to disrupt the bacterial membrane. In order to verify this hypothesis, we examined by scanning electron microscopy (SEM) analysis the morphological changes of *A. baumannii* ATCC 17978 when it was subjected to starvation for 24 h, followed by challenge with econazole, colistin, or combinations of the two agents. Upon starvation for 24 h, the cell boundary of *A. baumannii* ATCC 17978 became slightly wrinkled. When the strain was treated with colistin or econazole alone, cellular morphology became more distorted but not disrupted. Compared with treatment with a single agent, significant damage in cellular structure with a completely disrupted membrane characterized by large holes and leaked cytoplasm was observed when

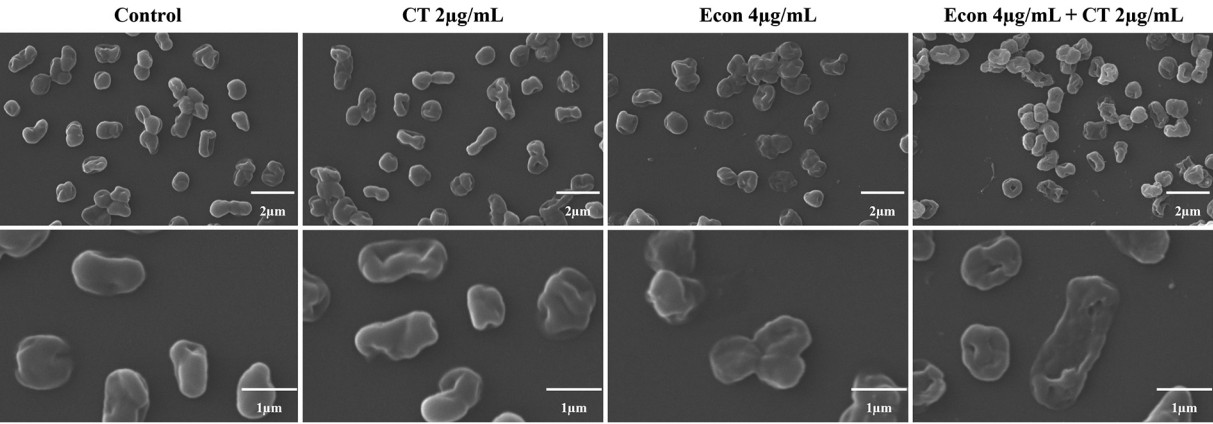

**FIG 5** Scanning electronic microscopy analysis of starvation-induced tolerant *A. baumannii* ATCC 17978. *A. baumannii* ATCC 17978 was subjected to starvation for 24 h and challenged with econazole, colistin, or combinations of the two agents for 4 h, followed by SEM analysis. Top panel: SEM MAG 20.0 kx; Bottom panel: SEM MAG 50.0 kx.

*A. baumannii* ATCC 17978 was challenged by the combination of colistin and econazole (Fig. 5).

To further investigate the ability of econazole to improve the membrane-damaging effect of colistin, we measured the changes in membrane permeability using SYTOX Green and in membrane potential using 3,3-dipropyl-thiadicarbocyanine [DiSC3(5)]. Both exponential and starvation-induced tolerant *A. baumannii* were tested. SYTOX Green is a fluorescent nucleotide dye that can stain the chromosome when the bacterial membrane becomes permeable due to structural damage. In exponentially growing *A. baumannii* (both ATCC 17978 and carbapenem-resistant CPC35), treatment with increased concentrations of econazole resulted in a slight increase in membrane permeability, but the degree of changes was low compared to that recorded upon treatment with colistin alone. However, combined treatment with 2 $\mu$g/mL colistin and various concentrations of econazole (2, 4, and 8 $\mu$g/mL) resulted in a significant increase in membrane permeability (Fig. 6a and b). On the other hand, *A. baumannii* bacteria which had encountered starvation stress for 24 h exhibited significant increase in membrane permeability compared to exponential-phase cells. Upon treatment with increased concentrations of econazole alone, the degree of bacterial membrane permeability increased further but was still lower than that recorded upon treatment with colistin alone. However, when challenged with a combination of 2 $\mu$g/mL colistin and econazole (2, 4, and 8 $\mu$g/mL), membrane permeability did not exhibit the dramatic increase observed in exponential phase, as the level recorded was similar to that caused by treatment with 2 $\mu$g/mL colistin alone (Fig. 6c and d). DiSC3(5), a cationic fluorescence dye commonly used for membrane potential determination, was employed to record the changes in membrane potential on both exponential and starvation-induced tolerant *A. baumannii* upon treatment with econazole, colistin, or combinations of the two agents, with valinomycin being used as a positive control. For both exponential-phase cells and the starvation-induced tolerant *A. baumannii* population, treatment with 2 $\mu$g/mL colistin alone did not cause much increase in membrane potential, whereas treatment with 4 $\mu$g/mL econazole alone resulted in a significant increase in membrane potential, which was even higher than that recorded upon treatment with 1 $\mu$g/mL valinomycin. Combined treatment with 2 $\mu$g/mL colistin and 4 $\mu$g/mL econazole also caused a significant increase in membrane potential, but the level was similar to that recorded during treatment with 4 $\mu$g/mL econazole alone (Fig. 7).

To further confirm the capability of econazole to enhance the bactericidal activity of colistin, sequential killing assays were performed on starvation-induced tolerant *A. baumannii* ATCC 17978. Upon starvation for 24 h, *A. baumannii* ATCC 17978 could not be killed when treated with econazole first, followed by treatment with colistin (Fig. 8a). However, when first treated with colistin alone for 3 h, starvation-induced

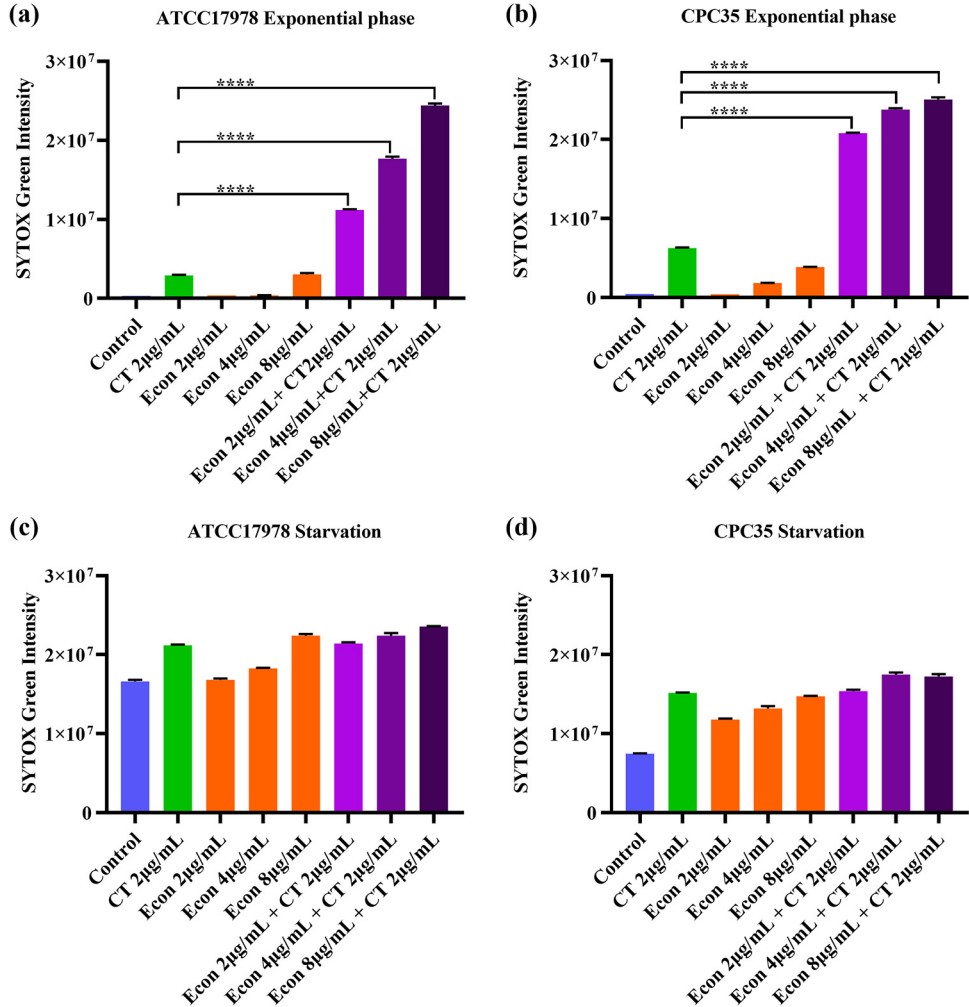

**FIG 6** Econazole caused an increase in membrane permeability in *A. baumannii*. Both exponential (a and b) and starvation-induced tolerant (c and d) *A. baumannii* strains ATCC 17978 (a and c) and CPC35 (b and d) were challenged with econazole, colistin, or combinations of the two agents for 4 h, followed by determination of membrane permeability using SYTOX Green. Fluorescence intensity was measured, and statistical analysis was conducted by one-way ANOVA and the Tukey test (****, $P < 0.0001$).

tolerant *A. baumannii* ATCC 17978 could be completely eradicated at 24 h upon exposure to econazole (Fig. 8b). These results suggested that treatment with colistin first disrupts the cell membrane of the *A. baumannii* strain and that further exposure to econazole causes PMF dissipation and eventually cell death.

## DISCUSSION

*Acinetobacter baumannii* is an important opportunistic pathogen that causes serious health care-associated infections with high morbidity and mortality in clinical settings (8). Treatment options for diseases caused by *A. baumannii* have become increasingly limited as a result of the emergence of multidrug-resistant strains, including those resistant to carbapenems, leaving polymyxins as the last therapeutic option. A variety of studies have evaluated the bactericidal activity of polymyxins when they are used in combination with other antifungal agents. The combination of polymyxin B and miconazole was demonstrated to exhibit synergistic antibacterial and antifungal activity against Gram-negative bacteria (including *Escherichia coli* and *Pseudomonas aeruginosa*) and yeast, respectively (26). In addition, Kim et al. reported polymyxin B with antifungal ciclopirox as a new synergistic combination with activity against multidrug-resistant *A. baumannii* and *E. coli* (27). It was also reported that combination of

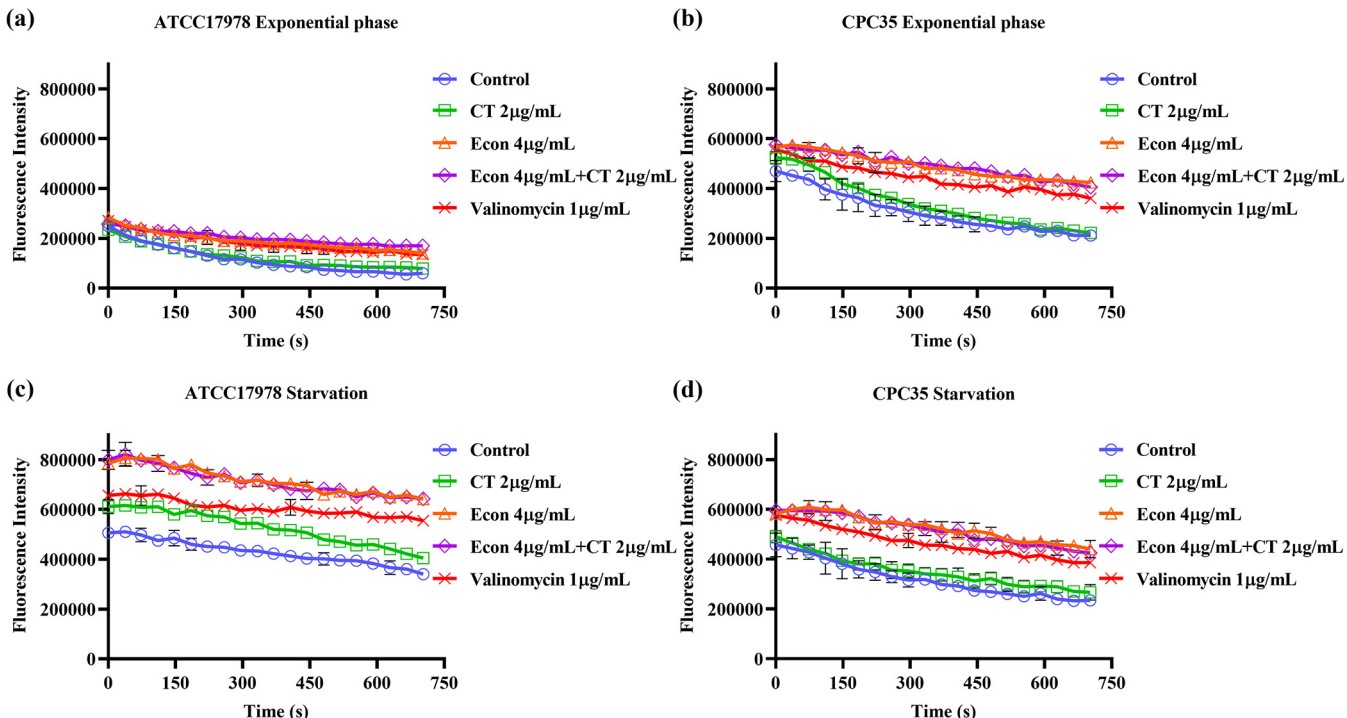

**FIG 7** Determination of changes in membrane potential of *A. baumannii*. Bacterial membrane potential was determined on both exponentially growing (a and b) and starvation-induced tolerant (c and d) *A. baumannii*. Fluorescent dye DiSC3(5) was employed to examine the changes in membrane potential upon treatment with colistin, econazole, or combinations of the two agents. Valinomycin (1 $\mu$g/mL) was used as a positive control.

polymyxin B and caspofungin showed significant synergistic antibacterial activity against multidrug-resistant (MDR) *K. pneumoniae* by inhibiting the bacterial virulence pathway and the multiresistant efflux mechanisms (28). A synergistic antimicrobial effect against multidrug-resistant *A. baumannii* could be illustrated when colistin was combined with azithromycin, minocycline, rifampin, vancomycin, or imipenem (29–32). Studies have also demonstrated that treatment with drug combinations that involve polymyxin B was more effective than that with polymyxin B alone, often reducing mortality in cases of multidrug-resistant *A. baumannii* infections (33, 34). Nevertheless, failures in drug combination therapy are still being reported, particularly in recalcitrance of chronic infections, caused by antibiotic-tolerant persister subpopulations (35). In fact, persisters may be a major cause of treatment failure and exhibit the potential to develop into antibiotic-resistant organisms. Therefore, eradication of persisters is imperative in managing both antibiotic tolerance and resistance in bacteria, and for

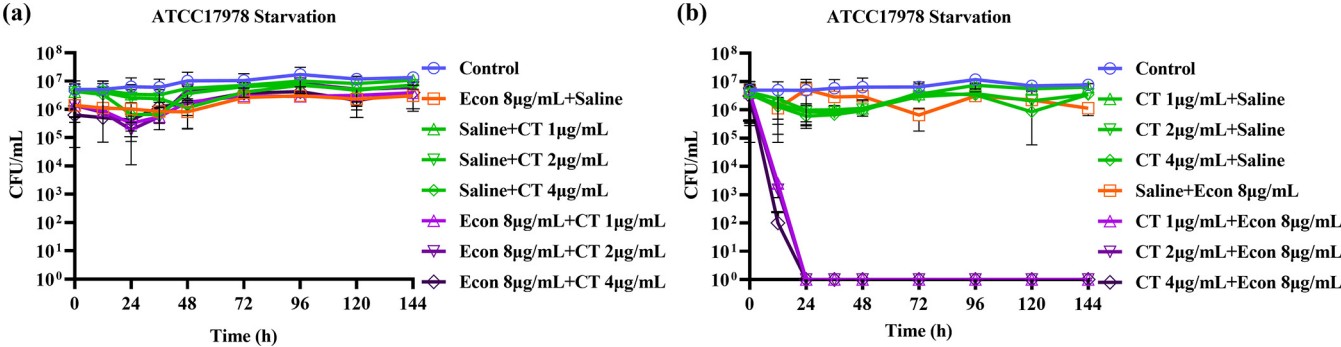

**FIG 8** Sequential killing of econazole and colistin on starvation-induced tolerant *A. baumannii* ATCC 17978. (a) Upon starvation for 24 h, strain ATCC 17978 was first treated with econazole for 3 h, followed by removal of econazole by washing twice with saline. *Coli*stin at the indicated concentration was then added. (b) Upon starvation for 24 h, strain ATCC 17978 was first treated with colistin for 3 h, followed by removal of colistin by washing twice with saline. Econazole at the indicated concentration was then added. Data were presented as mean $\pm$ standard deviation from three independent experiments.

developing new approaches to enhance treatment effectiveness and reduce suffering in patients. In this study, we evaluated the synergistic antimicrobial effect of econazole and colistin and showed that these two drugs can act in combination to eradicate the starvation-induced tolerant and multidrug-resistant populations of *A. baumannii*.

Findings in this work suggest that targeting the bacterial membrane would be an effective way to eradicate persisters (36). Colistin is known to cause membrane permeabilization in Gram-negative bacteria (37). The mechanism of action involves interaction with the outer membrane of Gram-negative bacteria and competitive displacement of $Mg^{2+}$ and $Ca^{2+}$, which stabilize the lipopolysaccharide layer, resulting in disruption of the membrane integrity (38). These events lead to the better penetration of econazole in colistin-treated cells. We recently showed that econazole itself can cause PMF dissipation and therefore enhances the damaging effects of colistin and promotes further penetration by both agents (25). PMF is interpreted as the sum of pH difference and membrane potential over bacterial membrane. Combination of colistin and tetrandrine showed synergistic antibacterial activity against colistin-resistant *Salmonella* in which tetrandrine enhanced the membrane-damaging ability of colistin by undermining the functions of PMF and efflux pumps (39). Econazole was recently reported to act in combination with conventional antibiotics including ampicillin, ceftazidime, gentamicin, and meropenem to effectively eradicate tolerant bacterial cells by dissipation of transmembrane PMF (40). Such a killing mechanism, which was not dependent on the bacterial growth status, may explain why combined use of the two agents results in effective eradication of nonreplicating persisters. Consistently, we found that the addition of econazole can significantly reduce the dose of colistin required to kill the tolerant and multidrug-resistant *A. baumannii* populations. As a major reason for restriction of clinical applications of colistin is a concern about its neurotoxicity and nephrotoxicity (41), a new approach to enhance the efficacy of colistin would allow the use of a much smaller dosage of the drug in clinical treatment, thereby drastically reducing its toxic side effects.

The limitations of the present study are that the synergistic antimicrobial activity of colistin and econazole combination was demonstrated on only a few clinical *A. baumannii* strains. Actually, we also tested the synergistic antimicrobial effects of colistin and econazole combination on another two clinical carbapenem-resistant *A. baumannii* strains, and they exhibited similar results in checkerboard assay, time-dependent killing assay, and membrane permeability assay (data not shown) as those observed in strain CPC35. Testing the antimicrobial activity of this drug combination on a larger number of bacterial strains would make the findings more convincing. Clinical trials are also required to support the application of this drug combination in practical situations. In conclusion, this study demonstrated that combination of colistin and econazole is effective in eradicating multidrug-resistant *A. baumannii* and its persisters. It is highly likely that this drug combination is also effective on other Gram-negative bacterial pathogens. Overall, this study emphasized the potential of repurposing econazole and provided a prospective alternative to the present limited usage of colistin in treating Gram-negative bacterial infections, which would eventually aid in clinical translation and assist in maintaining the activity of last-line polymyxins for the therapy of multidrug-resistant infections. Findings in this study therefore have important clinical implications and deserve further investigations to enhance their clinical application potential.

## MATERIALS AND METHODS

**Bacterial strains, drugs, and antimicrobial susceptibility test.** The reference strain *Acinetobacter baumannii* ATCC 17978 was purchased from the American Type Culture Collection (Manassas, VA). The clinical strain *A. baumannii* CPC35 was collected from a local hospital in Hong Kong and identified with a matrix-assisted laser desorption ionization–time of flight mass spectrometry (MALDI-TOF MS) apparatus (Bruker, Germany). Colistin sulfate salt (CAS no. 1264-72-8) and econazole nitrate salt (CAS no. 24169-02-6) were purchased from Sigma-Aldrich (USA). MIC of colistin alone or in combination with econazole was determined using the broth microdilution method recommended by the Clinical and Laboratory Standards Institute (CLSI) (42). *Escherichia coli* ATCC 25922 was used as a quality control. Synergistic antimicrobial effect of econazole and colistin combination was evaluated by a checkerboard assay (43), in

**TABLE 1** MICs of antibiotics against *A. baumannii* strains ATCC 17978 and CPC35

| Strain | MIC ($\mu$g/mL) of drug[a]: | | | | | | | | | | | |
|---|---|---|---|---|---|---|---|---|---|---|---|---|
| | AMP | AMK | GEN | TET | CTX | CRO | CT | CIP | NAL | CHL | IMP | MEM |
| ATCC 17978 | 16 | 2 | 2 | 1 | 8 | 8 | 4 | <0.25 | 16 | 64 | 0.12 | 0.25 |
| CPC35 | >128 | <8 | 8 | >64 | >128 | >128 | 2 | >8 | >128 | 32 | 32 | 32 |

[a]AMP, ampicillin; AMK, amikacin; GEN, gentamicin; TET, tetracycline; CTX, cefotaxime; CRO, ceftriaxone; CT, colistin; CIP, ciprofloxacin; NAL, nalidixic acid; CHL, chloramphenicol; IMP, imipenem; MEM, meropenem.

which effects of 2-fold serial dilutions of drug positioned in a 12 × 8 matrix were tested. After 18 h of incubation, absorbance of bacterial culture at 600 nm was determined using a microplate reader. At least three tests were conducted for each drug combination, and the means were used for fractional inhibitory concentration (FIC) calculation. The FIC for each drug was calculated by dividing the MIC of drug in combination by the MIC of that drug alone. FIC index (FICI) is the sum of the two FICs, with an FICI of ≤0.5 deemed synergistic, an FICI between 0.5 and 4 defined as indifferent, and an FICI of >4 deemed antagonistic. Strain CPC35 was resistant to imipenem, meropenem, cefotaxime, ceftriaxone, ciprofloxacin, nalidixic acid, and tetracycline (Table 1) and was defined as a multidrug-resistant *A. baumannii* strain.

**Time-dependent killing assay.** To generate an exponential-phase population, an overnight culture of *A. baumannii* was diluted in fresh Luria-Bertani (LB) broth and incubated for approximately 2 h to reach the exponential phase. Bacterial culture was then challenged with econazole, colistin, or various combinations of the two drugs at 37°C with constant shaking at 250 rpm. Viable counts were determined at 0, 1, 2, 3, 4, 6, and 8 h. To generate an antibiotic-tolerant population, a bacterial culture of exponential-phase cells was washed twice with saline (0.85% NaCl) and resuspended in saline, followed by incubation at 37°C with constant shaking at 250 rpm for 24 h to create starvation stress. Upon being starved for 24 h, the bacterial culture was then challenged with econazole, colistin, or various combinations of the two agents at 37°C with constant shaking. Viable counts were determined at 4, 24, 72, 120, and 144 h. Killing curves were depicted by plotting viable counts against time with GraphPad Prism 8 (San Diego, CA).

**Assessment of synergistic antimicrobial effect of econazole and colistin combination by LIVE/DEAD staining.** A LIVE/DEAD BacLight bacterial viability kit was employed to investigate the synergistic bactericidal effect of econazole and colistin combination against a starvation-induced antibiotic-tolerant *A. baumannii* subpopulation. Briefly, upon starvation for 24 h, bacterial cultures were treated with econazole, colistin, or different combinations of the two agents for 4 h, followed by washing twice with phosphate-buffered saline (PBS). Bacterial pellets were then resuspended in 100 $\mu$L PBS. LIVE/DEAD dye was added in the bacterial suspension, which was then incubated at 25°C for 15 min in the dark. After staining, bacterial suspensions were washed twice to remove excess dye and then resuspended in PBS. Images were captured using a Nikon Eclipse Ti2 microscope (Nikon, Tokyo, Japan). Quantitative analysis was performed by counting the number of live and dead cells. Statistical analysis was performed using one-way analysis of variance (ANOVA) and the Tukey test.

**Mouse sepsis infection model.** A mouse sepsis model was used as previously described (25) to assess the effectiveness of econazole and colistin combination in treatment of *A. baumannii* infection *in vivo*. ICR mice (4 weeks old, weighing 30 g) purchased from the Laboratory Animal Research Unit of the City University of Hong Kong were used in animal tests. Mice were made neutropenic by injecting 150 mg/kg and 100 mg/kg cyclophosphamide 3 days and 1 day respectively, before infection. Mice ($n = 8$) randomly assigned into each group were infected with $1.0 \times 10^8$ CFU of clinical carbapenem-resistant (colistin-susceptible) *A. baumannii* AB94 by intravenous injection. Then, mice were subjected to antibiotic treatment via intraperitoneal injection every 12 h for a total of 72 h, including saline, 4 mg/kg econazole, 0.25 mg/kg colistin, and combination of 4 mg/kg econazole and 0.25 mg/kg colistin. Survival of mice in each group was observed for a total of 120 h. Animal sepsis experiments were performed twice to verify the data consistency. Survival curves were delineated with statistical analysis conducted by log rank (Mantel-Cox) test in GraphPad Prism 8.

**Mouse thigh infection model.** The mouse thigh infection model was used as previously described (40) with slight modifications. Briefly, ICR mice (4 weeks old, weight of 30 g) were made neutropenic by injecting 150 mg/kg and 100 mg/kg cyclophosphamide 3 days and 1 day respectively, before infection. An inoculum of $1.80 \times 10^7$ CFU of clinical carbapenem- and colistin-resistant *A. baumannii* R44 (colistin MIC was 32 $\mu$g/mL) was injected into both thighs of each test mouse. After infection for 24 h, mice in each group ($n = 5$) received antibiotic treatment by intraperitoneal injection at 12-h intervals for a total of 72 h. Drug regimens tested included 8 mg/kg econazole, 8 mg/kg colistin, and combination of 8 mg/kg econazole and 8 mg/kg colistin. Saline was included as a control. Mice in each group were then euthanized, and both infected thighs were aseptically homogenized, after which serial dilutions were spread on LB plates. Bacterial loads of strain R44 in each group were recorded, and data analysis was performed by one-way ANOVA and the Tukey test.

**Examination of cellular morphology by SEM.** Scanning electron microscopy (SEM) was employed to examine the morphological changes of *A. baumannii* ATCC 17978 upon challenge with colistin, econazole, or combination of the two agents upon starvation for 24 h. Briefly, bacteria subjected to starvation for 24 h were treated with test agents and incubated at 37°C for 4 h. After incubation, the bacterial culture was washed, fixed, dehydrated, and dried as previously reported (44). The bacteria were then

coated with gold, and cellular morphology was observed in a scanning electron microscope (Tescan Vega3).

**Membrane permeability assay.** Membrane permeability assays were performed on *A. baumannii* in both the exponential-phase population and the population under starvation for 24 h. The test organisms were challenged with econazole, colistin, or combinations of the two drugs by incubation at 37°C for 4 h. The bacterial culture was then washed twice and resuspended in PBS to a final concentration with an optical density at 600 nm ($OD_{600}$) of 0.2. SYTOX Green was added to the bacterial suspension until a final concentration of 1 $\mu$M was reached, followed by incubation for 15 min at room temperature in the dark. Fluorescence intensity was determined with a SpectraMax iD3 multimode microplate reader (Molecular Devices, USA) with excitation wavelength at 488 $\pm$ 10 nm and emission wavelength at 523 $\pm$ 10 nm (45). Statistical analysis was performed by one-way ANOVA and the Tukey test.

**Membrane potential assay.** The ability of econazole and colistin combination to induce dissipation of membrane potential in *A. baumannii* was measured using a voltage-sensitive dye, 3,3-dipropyl-thiadi-carbocyanine [DiSC3(5)] (46). *A. baumannii* at exponential phase or those bacteria subjected to starvation for 24 h were washed twice and resuspended in PBS supplemented wtih 0.1 M KCl to a final concentration with an $OD_{600}$ of 0.2. The bacterial suspension was then incubated with 1 $\mu$M DiSC3(5) at room temperature in the dark for 15 min, followed by treatment with econazole, colistin, or combinations of the two agents. Valinomycin (1 $\mu$g/mL) was used as a positive control. Fluorescence intensity was measured with a SpectraMax iD3 multimode microplate reader with excitation wavelength at 610 nm and emission wavelength at 660 nm.

**Sequential killing assay.** An overnight culture of *A. baumannii* was diluted in fresh LB broth and then incubated for approximately 2 h to reach exponential phase. The bacterial culture was then washed twice and resuspended in saline, followed by incubation at 37°C with constant shaking at 250 rpm for 24 h to create starvation stress. Upon starvation for 24 h, the bacterial culture was first treated with econazole or colistin for 3 h, followed by removal of the agent by washing twice in saline. The other agent at the indicated concentration was then added. The bacterial culture was incubated at 37°C with shaking, and viable counts were recorded every 24 h for a total of 144 h.

**Ethics statement.** Animal infection experiments were approved by the Research Animal Care and Use Committee of the City University of Hong Kong.

**Data availability.** All data associated with this study are available upon request.

## ACKNOWLEDGMENTS

This research was funded by the Research Impact Fund of Hong Kong (R5011-18F).

M.X. performed the experiments and drafted the manuscript; K.C. helped with animal experiments; E.W.-C.C. designed the experiments and edited the manuscript; S.C. supervised the whole project and edited the manuscript.

We declare no competing interests.

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
