## [Reviewer comments · Microbiology Spectrum]

Microbiology Spectrum

Synergistic antimicrobial effect of colistin in combination with econazole against multidrug-resistant *Acinetobacter baumannii* and its persisters

Miaomiao XIE, Kaichao Chen, Edward Chan, and Sheng Chen

Corresponding Author(s): Sheng Chen, CityU of Hong Kong

Review Timeline:

Submission Date:	March 13, 2022
Editorial Decision:	April 4, 2022
Revision Received:	April 14, 2022
Accepted:	April 15, 2022

Editor: Aude Ferran

Reviewer(s): Disclosure of reviewer identity is with reference to reviewer comments included in decision letter(s). The following individuals involved in review of your submission have agreed to reveal their identity: Samuele Sabbatini (Reviewer #2)

Transaction Report:

DOI: <https://doi.org/10.1128/spectrum.00937-22>

April 4, 2022

Prof. Sheng Chen
CityU of Hong Kong
Department of Infectious Diseases and Public Health
Department of PH
Kowloon
Hong Kong

Re: Spectrum00937-22 (Synergistic antimicrobial effect of colistin in combination with econazole against multidrug-resistant *Acinetobacter baumannii* and its persisters)

Dear Prof. Sheng Chen:

Thank you for submitting your manuscript to Microbiology Spectrum. Your article was reviewed and some modifications were suggested. You will find the reviewer's comments below.

Link Not Available

Sincerely,

Aude Ferran

Journals Department
Reviewer comments:

Reviewer #1 (Comments for the Author):

A very nice study. Some comments and suggestions as below:

Materials and Methods

- Suggest to include more information on the antibiotic and antifungal. For example the salt form, brand and where they were

purchased including their batch number.

- Suggest to indicate whether colistin or CMS (colistin methanesulfonate) was used.
- Suggest to include a statement on antibiotic and antifungal stability as samples were collected for viable counts at 72, 120 and 144h (for the antibiotic tolerant population studies).
- Suggest to do ANOVA for the mouse thigh infection model.

Discussion

- The Discussion is a bit too brief. Suggest to further expand and cite more relevant studies. For example, has there been any other studies that have tested other antifungals with polymyxins?
- Suggest to comment and include studies on other antibiotic or antibiotic combinations affecting PMF.
- Suggest to include the translation of results in terms of use of colistin and CMS (i.e. what is commonly used in practice).
- Suggest to discuss the concentrations and doses used in the studies.

Minor comments:

- Line 74: Suggest to review sentence and revise for clarity.
- Line 81-82: Suggest to review sentence and further expand on "chronic and recurrent". Suggest to avoid the term "manage" for these studies.
- Line 296, 298, 309, 532: Suggest to change "totally" to "a total of".
- Line 309: Suggest a full stop rather than a comma (between 72hrs and drug regimens).
- Line 313: To change performing to performed

Reviewer #2 (Comments for the Author):

The manuscript by Xie M. et al. describes the effect of a new antibiotic-antifungal combination for the treatment of multi-drug resistant *Acinetobacter baumannii* infections. The significance of the results is very high and they support the conclusions made by the authors. The methodology is accurate and correctly applied. To the best of my knowledge this is the first study that analyses the effect of econazole in combination on multi-drug resistant microorganisms and this is remarkable since the research on new drugs against this difficult-to-treat infections usually takes long time. Furthermore the interest on testing combinations of pre-existing drugs is gaining space in microbiology research since new discovered one, due to the high incidence of antimicrobial resistance onset, are not usually used alone or it is granted for a short time.

The manuscript is well and clearly written but there are some issues to be addressed.

Results

- In the checkerboard assay results, the authors must include something to confirm the synergistic effect of drugs. It could be sufficient to calculate some indexes (e.g. FICI) or use some software that are dedicated to.
 - Figure 1: The figure need to be improved. The authors have to include a description in figure legends regarding the meaning of colour boxes used. Even if intuitive, there could be errors in the interpretation of data. For example, the authors state that ATCC 17978 strain has MIC of 2 µg/mL for colistin even if the graph reports a light-coloured box in the corresponding space indicating bacterial growth. Maybe a legend describing what the colours correspond could be helpful in understanding the degree of growth inhibition caused by drugs.
 - Figure 3: The authors stated that Student's t test was used to analyse the data. A test for multiple comparisons like ANOVA or Kruskal wallis to find significance between treatments should be applied.
 - In in vivo models of infection, why two different strains of *A. baumannii* respect to the ones used in vitro were used? Why the authors chose the colistin-susceptible strain for sepsis model and the colistin-resistant one for thigh infection?
- As regard drugs concentrations used, did the authors know what is the serum peak concentration achieved by the schedule they used? This could be very important to state that colistin is used at lower concentrations in order to avoid toxicity.
- As above, in thigh infection model, since the drugs concentrations IP injected are even much higher than sepsis model, are the authors sure that colistin serum concentration did not achieve too much high level? Is it ok that tissue colistin concentration could be relatively higher than into serum, but the system must be controlled.
- There aren't references about in vivo models, are they totally new designed? Reference methods could be useful to justify these concerns.
- Figure 4: As above, statistical analyses with Student's t test is not correct.
 - Figure 6: Statistical analyses with Student's t test is not correct
 - Please check all figure legends for consistency in statistical methods description and application, number of experiments performed with technical replicates.

Discussion

- The authors must include limitations of the study. Use of a much higher number of bacterial strain in selected experiments could confirm these very interesting findings.
- Have the authors think about the possibility to test this combination against *A. baumannii* biofilm? They used a starvation-induced model that could be quite similar but it does not account for possible structure-induced resistances against the treatment.

Material and methods

- Statistical analyses section is missing. Please add a detailed description of methods used.

References

- Some references are wrongly stated.

Staff Comments:

Preparing Revision Guidelines

Please return the manuscript within 60 days; if you cannot complete the modification within this time period, please contact me. If you do not wish to modify the manuscript and prefer to submit it to another journal, please notify me of your decision immediately so that the manuscript may be formally withdrawn from consideration by Microbiology Spectrum.

Reviewer comments

The manuscript by Xie M. *et al.* describes the effect of a new antibiotic-antifungal combination for the treatment of multi-drug resistant *Acinetobacter baumannii* infections. The significance of the results is very high and they support the conclusions made by the authors. The methodology is accurate and correctly applied. To the best of my knowledge this is the first study that analyses the effect of econazole in combination on multi-drug resistant microorganisms and this is remarkable since the research on new drugs against this difficult-to-treat infections usually takes long time. Furthermore the interest on testing combinations of pre-existing drugs is gaining space in microbiology research since new discovered one, due to the high incidence of antimicrobial resistance onset, are not usually used alone or it is granted for a short time.

The manuscript is well and clearly written but there are some issues to be addressed.

Results

- In the checkerboard assay results, the authors must include something to confirm the synergistic effect of drugs. It could be sufficient to calculate some indexes (e.g. FICI) or use some software that are dedicated to.
- Figure 1: The figure need to be improved. The authors have to include a description in figure legends regarding the meaning of colour boxes used. Even if intuitive, there could be errors in the interpretation of data. For example, the authors state that ATCC 17978 strain has MIC of 2 µg/mL for colistin even if the graph reports a light-coloured box in the corresponding space indicating bacterial growth. Maybe a legend describing what the colours correspond could be helpful in understanding the degree of growth inhibition caused by drugs.
- Figure 3: The authors stated that Student's t test was used to analyse the data. A test for multiple comparisons like ANOVA or Kruskal wallis to find significance between treatments should be applied.
- In *in vivo* models of infection, why two different strains of *A. baumannii* respect to the ones used *in vitro* were used? Why the authors chose the colistin-susceptible strain for sepsis model and the colistin-resistant one for thigh infection?
As regard drugs concentrations used, did the authors know what is the serum peak concentration achieved by the schedule they used? This could be very important to state that colistin in used at lower concentrations in order to avoid toxicity.
As above, in thigh infection model, since the drugs concentrations IP injected are even much higher than sepsis model, are the authors sure that colistin serum concentration did not achieve too much high level? Is it ok that tissue colistin concentration could be relatively higher than into serum, but the system must be controlled.
There aren't references about *in vivo* models, are they totally new designed? Reference methods could be useful to justify these concerns.
- Figure 4: As above, statistical analyses with Student's t test is not correct.
- Figure 6: Statistical analyses with Student's t test is not correct
- Please check all figure legends for consistency in statistical methods description and application, number of experiments performed with technical replicates.

Discussion

- The authors must include limitations of the study. Use of a much higher number of bacterial strain in selected experiments could confirm these very interesting findings.
- Have the authors think about the possibility to test this combination against *A. baumannii* biofilm? They used a starvation-induced model that could be quite similar but it does not account for possible structure-induced resistances against the treatment.

Material and methods

- Statistical analyses section is missing. Please add a detailed description of methods used.

References

- Some references are wrongly stated.

Reviewer comments:

Reviewer #1 (Comments for the Author):

A very nice study. Some comments and suggestions as below:

Materials and Methods

1. Suggest to include more information on the antibiotic and antifungal. For example, the salt form, brand and where they were purchased including their batch number.

Response: "Colistin sulfate salt (CAS-No.: 1264-72-8) and econazole nitrate salt (CAS-No.: 24169-02-6) were purchased from Sigma-Aldrich (USA)." was added to the Methods.

2. Suggest to indicate whether colistin or CMS (colistin methanesulfonate) was used.

Response: Colistin sulfate salt was used.

3. Suggest to include a statement on antibiotic and antifungal stability as samples were collected for viable counts at 72, 120 and 144h (for the antibiotic tolerant population studies).

Response: We have not tested the stability of colistin and econazole in our assay condition. But data from some of our assays showed that the population of persisters decreased gradually even at 72 hours suggesting that these compounds are active at this time point. In addition, we also performed the assay by adding colistin and econazole every two days and got similar results.

4. Suggest to do ANOVA for the mouse thigh infection model.

Response: Statistical analysis were performed by one-way ANOVA and Tukey test in mouse thigh infection model. And P values between groups were re-calculated.

Discussion

5. The Discussion is a bit too brief. Suggest to further expand and cite more relevant studies. For example, has there been any other studies that have tested other antifungals with polymyxins?

Response: "The combination of polymyxin B and miconazole was demonstrated to exhibit synergistic antibacterial and antifungal activity against Gram-negative

bacteria (including Escherichia coli and Pseudomonas aeruginosa) and yeast, respectively. In addition, Kim et al reported polymyxin B with antifungal ciclopirox as a new synergistic combination with the activity against multidrug-resistant A. baumannii and E. coli. It was also reported that combination of polymyxin B and caspofungin showed significant synergistic antibacterial activity against MDR K. pneumoniae by inhibiting the bacterial virulence pathway and the multi-resistant efflux mechanisms.” have been added to Discussion.

6. Suggest to comment and include studies on other antibiotic or antibiotic combinations affecting PMF.

Response: “PMF is interpreted as the sum of pH difference and membrane potential over bacterial membrane. Combination of colistin and tetrandrine showed synergistic antibacterial activity against colistin-resistant Salmonella in which tetrandrine enhanced the membrane-damaging ability of colistin by undermining the functions of PMF and efflux pumps. Econazole was recently reported to act in combination with conventional antibiotics including ampicillin, ceftazidime, gentamicin and meropenem to effectively eradicate tolerant bacterial cells by dissipation of transmembrane PMF.” have been added to Discussion.

7. Suggest to include the translation of results in terms of use of colistin and CMS (i.e. what is commonly used in practice).

Response: “Overall, this study emphasized the potential of repurposing econazole and provided a prospective alternative to the present limited usage of colistin in treating Gram-negative bacterial infections, which would eventually aid in the clinical translation and assist in maintaining the activity of last-line polymyxins for the therapy of multidrug-resistant infections.” have been added to Discussion.

8. Suggest to discuss the concentrations and doses used in the studies.

Response: The doses used was firstly determined by MIC of the combination of colistin and econazole. The MIC value in the unit of mg/kg of mice will be used as the first point treatment dose. Two to five folds of MIC concentrations will then be used to get the best treatment doses.

Minor comments:

9. Line 74: Suggest to review sentence and revise for clarity.

Response: This sentence has been revised to "Treatment with a combination of antibiotics has also been proposed as an effective way to eradicate persisters that cause chronic and recurrent infections."

10. Line 81-82: Suggest to review sentence and further expand on "chronic and recurrent". Suggest to avoid the term "manage" for these studies.

Response: This sentence has been revised to "Our preliminary data show that this drug combination exhibited strong antimicrobial activity against these difficult-to-treat A. baumannii infections."

11. Line 296, 298, 309, 532: Suggest to change "totally" to "a total of".

Response: Revised accordingly.

12. Line 309: Suggest a full stop rather than a comma (between 72hrs and drug regimens).

Response: Revised accordingly.

13. Line 313: To change performing to performed

Response: Revised accordingly.

Reviewer #2 (Comments for the Author):

The manuscript by Xie M. et al. describes the effect of a new antibiotic-antifungal combination for the treatment of multi-drug resistant *Acinetobacter baumannii* infections. The significance of the results is very high and they support the conclusions made by the authors. The methodology is accurate and correctly applied. To the best of my knowledge this is the first study that analyses the effect of econazole in combination on multi-drug resistant microorganisms and this is remarkable since the research on new drugs against this difficult-to-treat infections usually takes long time. Furthermore, the interest on testing combinations of pre-existing drugs is gaining space in microbiology research since new discovered one, due to the high incidence of antimicrobial resistance onset, are not usually used alone or it is granted for a short time.

The manuscript is well and clearly written but there are some issues to be addressed.

Results

1. In the checkerboard assay results, the authors must include something to confirm the synergistic effect of drugs. It could be sufficient to calculate some indexes (e.g., FICI) or use some software that are dedicated to.

Response: “and the means were used for fractional inhibitory concentration (FIC) calculation. The FIC for each drug was calculated by dividing the MIC of drug in combination by the MIC of that drug alone. FIC index (FICI) is the sum of the two FICs, with an FICI ≤ 0.5 deemed synergistic, an FICI between 0.5 and 4 defined as indifference and an FICI > 4 deemed antagonistic.” were added to the Methods.

FIC indexes were calculated for strains ATCC 17978 and CPC35 and added to the Results.

2. Figure 1: The figure needs to be improved. The authors have to include a description in figure legends regarding the meaning of color boxes used. Even if intuitive, there could be errors in the interpretation of data. For example, the authors state that ATCC 17978 strain has MIC of 2 $\mu\text{g}/\text{mL}$ for colistin even if the graph reports a light-colored box in the corresponding space indicating bacterial growth. Maybe a legend describing what the colors correspond could be helpful in understanding the degree of growth inhibition caused by drugs.

Response: Dark blue regions represent higher cell density.

Colistin MIC of strain ATCC 17978 has been revised to 4 $\mu\text{g}/\text{mL}$.

3. Figure 3: The authors stated that Student's t test was used to analyse the data. A test for multiple comparisons like ANOVA or Kruskal wallis to find significance between treatments should be applied.

Response: Data were analyzed using one-way ANOVA and Tukey test. P values between groups were re-calculated.

4. In in vivo models of infection, why two different strains of *A. baumannii* respect to the ones used in vitro were used? Why the authors chose the colistin-susceptible strain for sepsis model and the colistin-resistant one for thigh infection?

Response: Because both strains ATCC17978 and CPC35 could not cause the mice

death even mice were infected with a very high inoculum (such as 4.0×10^8 CFU) in mouse sepsis model, we used a more virulent clinical strain AB94.

We first used colistin-susceptible strain AB94 in mouse thigh model, but results showed that both colistin single treatment and colistin/econazole combined treatment were very effective against the colistin-susceptible strain AB94 and there have no significant differences between these two groups, so we used a colistin-resistant strain R44 in mouse thigh model.

5. As regard drugs concentrations used, did the authors know what is the serum peak concentration achieved by the schedule they used? This could be very important to state that colistin is used at lower concentrations in order to avoid toxicity.

Response: We have not performed the pK/pD study for this combination yet. The statement for the lower toxicity is based on the in vivo efficacy data. At the effective dose of colistin in combination treatment, colistin alone did not show efficacy on the treatment of infections caused by A. baumannii, which suggested that higher dose will be needed when colistin is used alone for treatment. Therefore, the use of lower dose of colistin in combination suggested the lower toxicity when compared to single use of colistin.

6. As above, in thigh infection model, since the drugs concentrations IP injected are even much higher than sepsis model, are the authors sure that colistin serum concentration did not achieve too much high level? Is it ok that tissue colistin concentration could be relatively higher than into serum, but the system must be controlled.

Response: For sepsis model, the treatment was started right after the inoculation of bacteria. However, for thigh infection model, we tried to create a persister model by inoculating the bacteria, but treatment started 24 hours after the inoculation. During these 24 hours, most of the bacteria inoculated supposed to become persister in mice. Therefore, we need to use higher dose for treatment for this persister model. In addition, 8mg/kg dose of colistin is not high for mice, which has been used in our other study. Furthermore, we also did a maximum tolerance dose for colistin and showed that combination (80mg/kg econazole + 32mg/kg colistin) injection for six consecutive days did not produce any visible signs of illness in the test mice. (Xu C, Chen K, Chan KF, Chan EWC, Guo X, Chow HY, Zhao G, Zeng P, Wang M, Zhu Y.

2020. Imidazole Type Antifungal Drugs Are Effective Colistin Adjuvants That Resensitize Colistin-Resistant Enterobacteriaceae. *Advanced Therapeutics* 3:2000084.)

7. There aren't references about in vivo models, are they totally new designed? Reference methods could be useful to justify these concerns.

Response: Reference "Xu C, Chen K, Chan KF, Chan EWC, Guo X, Chow HY, Zhao G, Zeng P, Wang M, Zhu Y. 2020. Imidazole Type Antifungal Drugs Are Effective Colistin Adjuvants That Resensitize Colistin-Resistant Enterobacteriaceae. Advanced Therapeutics 3:2000084." was added to mouse sepsis infection model.

Reference "Wang M, Chan EWC, Xu C, Chen K, Yang C, Chen S. 2022. Econazole as adjuvant to conventional antibiotics is able to eradicate starvation-induced tolerant bacteria by causing proton motive force dissipation. Journal of Antimicrobial Chemotherapy 77:425-432." was added to mouse thigh infection model.

8. Figure 4: As above, statistical analyses with Student's t test is not correct.

Response: Statistical analysis were performed by one-way ANOVA and Tukey test in mouse thigh infection model. And P values between groups were re-calculated.

9. Figure 6: Statistical analyses with Student's t test is not correct

Response: Statistical analysis were performed by one-way ANOVA and Tukey test in membrane permeability assays. And P values between groups were re-calculated.

10. Please check all figure legends for consistency in statistical methods description and application, number of experiments performed with technical replicates.

Response: Statistical methods description and application as well as number of experiments performed with replicates have been checked for all figures.

Discussion

11. The authors must include limitations of the study. Use of a much higher number of bacterial strains in selected experiments could confirm these very interesting findings.

Response: "The limitations of present study are that the synergistic antimicrobial activity of colistin and econazole combination was demonstrated on only few clinical A. baumannii strains. Actually, we also tested the synergistic antimicrobial effects of

colistin and econazole combination on other two clinical carbapenem-resistant A. baumannii strains and they exhibited similar results in checkerboard assay, time-dependent killing assay and membrane permeability assay (data not shown) as observed in strain CPC35. Testing the antimicrobial activity of this drug combination on a larger number of bacterial strains would make the findings more convincing. Clinical trials are also required to support the application of this drug combination in practical situations.” have been added to Discussion.

Results of other two clinical carbapenem-resistant A. baumannii strains CPC37 and CPC52 in (a)-(b) checkerboard assays, (c)-(f) time-dependent killing assays and (g)-(j) membrane permeability assays are shown as follows (see figures below). Both strains CPC37 and CPC52 exhibited similar results in all these three assays as observed by strain CPC35, so results of strains CPC37 and CPC52 were not included in the manuscript.

12. Have the authors think about the possibility to test this combination against *A. baumannii* biofilm? They used a starvation-induced model that could be quite similar, but it does not account for possible structure-induced resistances against the

treatment.

Response: We performed the biofilm growth and treatment assays to test the ability of colistin and econazole combination against A. baumannii biofilm. And results showed that combination of colistin and econazole did not exhibit synergistic anti-biofilm activity against A. baumannii strains.

Material and methods

13. Statistical analyses section is missing. Please add a detailed description of methods used.

Response: Statistical analysis methods were added to corresponding assays.

References

14. Some references are wrongly stated.

Response: References have been checked and revised.

April 15, 2022

Prof. Sheng Chen
CityU of Hong Kong
Department of Infectious Diseases and Public Health
Department of PH
Kowloon
Hong Kong

Re: Spectrum00937-22R1 (Synergistic antimicrobial effect of colistin in combination with econazole against multidrug-resistant *Acinetobacter baumannii* and its persisters)

Dear Prof. Sheng Chen:

Thank you for your answers to reviewers and for the modifications made in the manuscript. Your manuscript has been accepted, and I am forwarding it to the ASM Journals Department for publication. You will be notified when your proofs are ready to be viewed.

Sincerely,

Aude Ferran
Editor, Microbiology Spectrum
